# From Quantum Materials to Microsystems

**DOI:** 10.3390/ma15134478

**Published:** 2022-06-25

**Authors:** Riccardo Bertacco, Giancarlo Panaccione, Silvia Picozzi

**Affiliations:** 1Dipartimento di Fisica, Politecnico di Milano, 20133 Milan, Italy; 2Istituto di Fotonica e Nanotecnologie CNR-IFN, 20133 Milan, Italy; 3Laboratorio TASC in Area Science Park—Basovizza, CNR-IOM, 34149 Trieste, Italy; giancarlo.panaccione@iom.cnr.it; 4Consiglio Nazionale delle Ricerche, CNR-SPIN c/o Università G. D’Annunzio, 66100 Chieti, Italy; silvia.picozzi@spin.cnr.it

**Keywords:** quantum materials, ferroics, microsystems

## Abstract

The expression “quantum materials” identifies materials whose properties “cannot be described in terms of semiclassical particles and low-level quantum mechanics”, i.e., where lattice, charge, spin and orbital degrees of freedom are strongly intertwined. Despite their intriguing and exotic properties, overall, they appear far away from the world of microsystems, i.e., micro-nano integrated devices, including electronic, optical, mechanical and biological components. With reference to ferroics, i.e., functional materials with ferromagnetic and/or ferroelectric order, possibly coupled to other degrees of freedom (such as lattice deformations and atomic distortions), here we address a fundamental question: “how can we bridge the gap between fundamental academic research focused on quantum materials and microsystems?”. Starting from the successful story of semiconductors, the aim of this paper is to design a roadmap towards the development of a novel technology platform for unconventional computing based on ferroic quantum materials. By describing the paradigmatic case of GeTe, the father compound of a new class of materials (ferroelectric Rashba semiconductors), we outline how an efficient integration among academic sectors and with industry, through a research pipeline going from microscopic modeling to device applications, can bring curiosity-driven discoveries to the level of CMOS compatible technology.

## 1. Introduction

### 1.1. Quantum Materials

Quantum materials identify a wide class of emergent materials whose properties cannot be approximated by macroscopic classical models as they rely on purely quantum mechanical mechanisms. Examples of quantum materials are superconductors, 2D materials, topological insulators, Rashba systems, Weyl semimetals, spin ices and quantum spin liquids. Notably, all of them display macroscopic properties arising from strong interactions between charge, spin, orbit and lattice degrees of freedom leading to collective excitations and quasi particles [1]. 

As such, quantum materials are nowadays at the cutting edge of materials research and test our ability to understand, design and control, i.e., ‘build’ systems with tailored functionalities, directly connected to emergent collective quantum phenomena [2,3]. Overall, the grand challenge at the heart of quantum materials is: how are strongly interacting electrons organized in matter, and how can this organization be controlled to implement novel functionalities? In particular, the perspective of creating ‘multi-functionality’ in micro/nano-electronics with ferroic materials has been a primary scientific and technological goal for more than a decade [4,5,6,7,8]. 

Properties to be combined/explored and probed, to name but a few, are the coexistence of long-range ferroelectric/magnetic order with the semiconducting character, room temperature operation, low-power and fast-switching by magnetoelectric coupling, tuning spin-orbit coupling (SOC) and dimensionality effects in tailored interfaces and nanostructures, manipulation of ferroelectric and magnetic domains by means of electric fields and/or by a single laser pulse, spin-charge interconversion in high SOC materials and topological materials, etc. [9,10,11]. Notably, when it comes to quantum materials and topological orders, the concept of order parameter is not the only concept to take into account, thus also requiring consideration of the geometric/symmetric property of the electron wave functions [12]. 

### 1.2. The Lessons from the “Gold Age” of the Quantum Era

Remarkably, in the introduction and conclusion of all papers focused on quantum materials, the authors sell their product with sentences such as: “this is crucial in view of applications to …”, “this paves the way to the development of a new technology platform …”. But how far are we from a real impact of quantum materials on technology, on economy developments and, eventually, on our society? No doubt any exploitation will involve microelectronic industry so that we could reduce the complexity of the previous questions by saying: “How far are quantum materials from being exploited in microsystems, including electronic, photonic and micro-electromechanical devices?”. Unfortunately, the answer is: “Quite far away”, at least for the vast majority of novel “quantum materials” currently under investigation. 

However, let us focus on the very beginning of the quantum era. During the last century, the revolution of quantum mechanics triggered an incredible race towards the understanding of materials, thanks to the development of solid-state physics, which immediately gave rise to the exploitation of these concepts in the semiconductor industry. 

The Bloch theorem, the quantum mechanics foundation of band structure in solids, dates back to 1928 [13]. Immediately after the Second World War, in 1948, Brattain, Bardeen and Shockley invented the bipolar junction transistor (BJT) at the Bell Labs. It was simply a laboratory prototype, but by 1954, the first portable transistor radios, typically using four to eight transistors, were already being released. In 1959, Mohamed Atalla and Dawon Kahng, again at Bell Labs, invented the metal-oxide semiconductor field effect transistor (MOSFET), the first compact transistor compatible with miniaturization, mass-production and integrated circuits. This was the beginning of the microelectronic industry. The number of transistors produced per year was less than 70 in 1960, but exponentially grew so that nowadays, a single microprocessor contains 57 billion transistors [14]. Remarkably, the impressive know-how developed in silicon microfabrication for the electronic industry has also been exploited in other applications such as microelectromechanical systems (MEMS), biosensors and other types of device that belong to the wide class of microsystems. 

What should we learn from the semiconductor story? *First*: one cannot skip relevant methodological steps. It took 20 years of hard and systematic work to go from the Bloch theorem to the first transistor and 10 more years to develop the foundations of CMOS technology. *Second*: good ideas become technologically relevant only thanks to a close partnership between academia and industry. That was the situation at Bell Labs: fundamental research carried out by eminent scientists in a laboratory of a large company, which is something that nowadays is almost completely missing, at least in Europe. Companies often tend to carry out internally, only very applied research, an activity mostly finalized to new products expected for the following year, i.e., with a brief time horizon, while the connection between academia and high-tech companies is not always smooth. That is the overall message: disruptive, non-incremental progress starts by addressing fundamental questions, but fundamental answers will never find a killer application if the scientific community that produced them does not interact with industry. In other words, an efficient integration between basic and industrial research seems the only key to real technological breakthroughs.

### 1.3. CMOS “Will Never Die”

The widely used argument by people working on novel, functional and sometimes exotic materials to motivate the effort on quantum materials is based on two assumptions: (i) we are approaching the limits of validity of Moore’s law stating that the density of transistors on a chip roughly doubles every two years and (ii) CMOS technology must be replaced by another technology to extend Moore’s law beyond the current limits [15,16]. The validity of the first assumptions has been questioned by many authors [17] and by recent announcements by IBM [18] of the “First 2 Nanometer Chip Technology”. The use of 3D packaging, refined materials engineering and defect control allow for device down-scaling and device areal densities which could not even be imagined a few years ago. All of this is based on the evolution of silicon-based CMOS technology, thanks to an enormous market-driven effort by semiconductor companies that have accumulated an impressive know-how on each scientific aspect behind CMOS technology: material science, device modeling, fabrication processes and system integration. This work has been carried out in about 50 years, since the invention of the complementary MOS (CMOS) process in 1963, by Chih-Tang Sah and Frank Wanlass at Fairchild Semiconductor. The result is a technology platform with a degree of maturity and economic impact which has no equal. That is why we can say that “CMOS will never die”. This said, there is a peculiarity with CMOS that we should not forget: CMOS is essentially exploited for digital applications, with transistors used as switches with two states, i.e., ON and OFF. For digital applications and Boolean computing, it is thus difficult to imagine that another platform can replace CMOS in the middle term. However, nowadays there is a renewed interest in analog applications, using a continuous variable of state instead of simply zeros and ones, for computing and signal processing. Coherence phenomena typical of light are being integrated into micro- and nano-scale devices in the so-called “wave computing” approach. Brain-inspired or neuromorphic approaches that exploit a completely different scheme with respect to the conventional Von Neumann one, are evolving towards a processor unit embedded in the memory unit (in-memory computing). Entanglement and other pure quantum phenomena are being used in q-bits in the field of quantum computing, now evolving from single atoms or photons towards macroscopic systems integrated onto silicon chips. In this framework, novel quantum materials play a fundamental role: not to replace silicon and CMOS, but to add new functionalities in the field of computing with the fundamental requirement of being “CMOS-compatible”.

### 1.4. Aim of the Paper

In this paper we analyze a specific class of quantum materials, i.e., ferroic materials, on which the authors have been working for the last 20 years, and we outline our roadmap towards the realistic exploitation of ferroics in microelectronic devices. The story of semiconductors shows that a rigorous methodology is needed to develop the long research pipeline, as summarized in Figure 1. First: theory and modeling, to unveil new microscopic mechanisms and emergent phenomena, shortlist new materials with desired functionalities and simulate their behavior from ideal “bulk-like” conditions to real devices. Second: material synthesis and advanced characterization. Beyond the “flake exfoliation and single crystal cleavage” approach, thin film and multilayer growth must be managed, with the benchmark of semiconductor standards in terms of purity, defect concentration, film roughness, stress control and uniformity over large areas compatible with device fabrication. The investigation of functional properties must be carried out using cutting-edge techniques with the fundamental requirement of reproducibility and accuracy for each measurement. The common approach “One measurement, one paper; two measurements, no paper”, essentially motivated by the anxiety of researchers connected to the value of their bibliometric indexes, is extremely dangerous for science and hopeless for the development of new technology platforms. Third: device design, fabrication and characterization. The “trial and error” approach is too demanding in terms of human and financial resources, while a good simulator can help the design and allow for a deeper understanding of prototype characterization. As above, reproducibility and accuracy of results must be strict requirements. Fourth: technology transfer. Prototype fabrication must be carried out with the clear perspective of process scalability and integration with CMOS technology: not a single microelectronic company will ever consider a process which cannot be scaled up to 12” wafer size and integrated with CMOS processes. Scientists must, therefore, be strongly connected to semiconductor companies in order to develop realistic technology platforms.

How to apply this methodology to the case of ferroic materials is described below. First, we recall the main characteristics of ferroics and stress on the unique opportunities offered by ferroics to build-up analog computing platforms that are complementary and integrated with CMOS digital processors. Second, we describe a roadmap on modeling, materials synthesis and characterization, device design, fabrication and characterization and exploitation. Our point of view is strengthened by describing the prototypical example of GeTe, where such a pipeline has been fruitfully exploited. Finally, we draw our conclusions.

## 2. (Multi-)Ferroic Materials

Quantum materials showing co-operative phenomena characterized by long-range ferroic order (i.e., ferroelectric, ferroelastic and (anti-)ferromagnetic) are extremely appealing in the abovementioned framework of adding new functionalities to CMOS-based electronics. Indeed, their order parameter (electric polarization, spontaneous deformation and magnetization for ferroelectrics, ferroelastics and ferromagnets, respectively) can be naturally exploited as a state macroscopic variable to encode multi-level memory states. As such, ferroic materials inherently represent excellent candidates for non-volatile memory devices. 

Despite the microscopic origin of the order parameter being inherently, different ferroic materials such as (anti-)ferroelectrics and (anti-)ferromagnets share many similarities. In particular, they all show: (i) ordering temperature (so-called Curie (Néel) temperature) below which a phase transition occurs and the long-range order sets in; (ii) hysteresis cycles in the polarization/electric-field phase space for ferroelectrics and in the magnetization/magnetic-field phase space for ferromagnets and (iii) domains, i.e., regions of the material where the order parameter is constant, with neighboring domains showing different direction and/or sign of the order parameter.

Several ferroics are actually “multiferroic” (i.e., they show more than one kind of long-range order, e.g., ferroelectricity and magnetism) and/or they are characterized by a remarkable cross-coupling between different degrees of freedom (spin, charge, lattice, etc.). In this context, piezoelectric, magnetostrictive and magnetoelectric materials (in bulk form or in composites) constitute a never ending source of rich physics and potentially promising applications. Indeed, the presence of more than one kind of active degrees of freedom enables the exploitation of multiple variables of state in the same device. In other words, it allows (i) multiple input-channels and conversion from one input to another output (appealing for computing devices) and (ii) high-sensitivity to various external stimuli (appealing for sensors).

Room temperature operation is often a requirement for applications: in many cases this goal is easily achieved (i.e., standard ferroelectric perovskites, simple metallic ferromagnets), but in some cases (i.e., multiferroic oxides) it still remains a challenge and an area where efforts should be directed in the coming years. 

As detailed below in Section 3, future perspectives of quantum ferroics are multifaceted. Thanks to the small values of the switching energy (a few fJ) displayed by ferroelectric and magnetoelectric materials [19,20], quantum ferroics can pave the way to low-power consumption, relevant in the context of “green electronics”. In case of tunable-switching energy barriers, the traditional paradigm of long retention times for conventional memories can shift to stochastic switching, an appealing functionality for stochastic computing. Another ever increasing area is the control over domain size, switching energy barriers and viscosity, which open the path to the exploitation of continuous values for the macroscopic state variables and plasticity in the framework of neuromorphic computing. Wave excitations of the ground state associated to a given ferroic order can be used for signal processing and wave computing with unique features in terms of non-linearity, low propagation losses and opportunities for quantum computing schemes associated to their bosonic nature. 

To better illustrate the potential and the state of the art of quantum ferroic materials, let us consider the specific case of applications in a field of unconventional computing, i.e., neuromorphic computing. In Table 1, we report a comparison between neuromorphic (analog) and CMOS (digital) approaches to the problem of writing/reading a variable of state which represents the synaptic weight in a neural network. This represents a fundamental block of all algorithms of patterning recognition based on neural networks, e.g., a perceptron, where the output of the n-th layer is the result of an activation function on a sum of the input from the (n − 1)-th layer with coefficients given by the synaptic weights [21]. As a benchmark, we consider the access to a 32-bit word in a DRAM [22] used as cache memory for storing the synaptic weights in a microcontroller which implements a digital version of the weighted sum. Concerning quantum ferroic materials, we report values for promising analog architectures exploiting a ferroic order parameter as a state variable for storing synaptic weights: (i) ferroelectric Rashba semiconductors (FERSC) involved in ferroelectric spin-orbit devices (FESO) [23] and (ii) ferroelectric two-dimensional electron gas at oxide interfaces (FE 2DEG) [24] in devices similar to the magneto-electric spin orbit device (MESO) proposed by INTEL [19], both exploiting the ferroelectric control of spin-to-charge conversion in Rashba systems to read a ferroelectric variable of state and (iii) topological insulators (TI) to enhance charge-to-spin conversion in spin orbit torque magnetic tunneling junctions (SOT MTJs) where the synaptic weight is written in the micromagnetic state of a ferromagnetic electrode [25]. The fundamental difference with respect to the digital solution is that the weighted sum is performed in the analog world by exploiting Kirchhoff’s laws and some memristive elements storing the synaptic weights in their actual impedance value, depending on the ferroic state variable, which can be set in the training phase. For FERSC (FESO), data are taken from ref [23] by scaling them to a target device size of 40 nm. For FE 2DEG systems, values are derived from results on devices based on SrTiO_3_ single crystals [24] and from BaTiO_3_-based ferroelectric tunneling junctions [20]. For SOT, MTJs state-of-the art values from devices exploiting heavy metals for SOT are reported [25], as the deployment of TI in this class of devices is still in its infancy.

Table 1 clearly demonstrates the potential of ferroic quantum materials, mainly in terms of high retention time (i.e., negligible steady state consumption for refreshing), and energy saving for accessing and processing a numerical value that could be stored in the transconductance of a device. On the other hand, the same table also provides evidence for the need of systematic and extensive work to bring these promising device concepts to the level of industrial maturity which is needed to compete with a well-established technology such as DRAMs. As a matter of fact, preliminary studies have been carried out only on small laboratory samples and only for some systems, e.g., FE 2DEG, the operation is still limited to cryogenic temperatures.

In the remainder of the paper, we outline a roadmap for making these concepts a realistic technology, bridging the gap from academic research to industrial microsystems.

## 3. The Research Pipeline in Quantum Ferroics

### 3.1. Modelling: The Importance of Microscopic Mechanisms

The first step to take along the quantum ferroics research pathway is theory and modelling, in turn subdivided into several avenues, as discussed below. The initial driving force is the identification of novel emergent microscopic mechanisms: in this respect, the field of ferroics is particularly rich and unexplored, as shown in the last two decades of tantalizing discoveries. To mention just one paradigmatic example in the field of multiferroics, in early 2000, the idea that, in frustrated magnets, such as TbMnO_3_ [26] and other manganites, complex exchange interactions (such as competing ferromagnetic and antiferromagnetic Heisenberg-like terms or antisymmetric Dzyaloshinskii-Moriya exchange or symmetric anisotropic exchange, etc.), giving rise to complex spin textures (such as spin spirals) could also give rise to ferroelectric polarization was put forward to explain the experimental observations in TbMnO_3_ and related compounds. There, the breaking of inversion symmetry induced by non-centrosymmetric magnetic ordering was found to produce improper electronic ferroelectricity, i.e., where the main order parameter of the phase transition is of magnetic origin, but it is one-to-one linked to ferroelectric polarization. In multiferroic manganites, the magnetoelectric coupling was found to be intrinsically strong since spin and electric dipolar orders share the same microscopic origin. In 2008, a Heisenberg exchange coupling was theoretically proposed to give rise to a relatively large ferroelectric polarization in frustrated collinear antiferromagnets, such as E-type HoMnO_3_, showing up-up-down-down spin chains [27,28]. This theory proposal, supported by first-principles simulations, led several experimental groups to probe and confirm the presence of ferroelectric polarization driven by the Heisenberg-like microscopic mechanism.

The abovementioned example remarkably shows that, when dealing with novel microscopic mechanisms, one needs not only intuition supported by a deep understanding of the underlying physics but also some tools that are able to quantify their effects and potential scientific and technological impact. The most popular modelling approach that seems capable to predict the strength and efficiency of microscopic mechanisms is density functional theory (DFT) [29]. First-principles simulations are indeed able to treat, on the same footing, all the ingredients at play, without being forced to select one or a few mechanisms in the initial modelling stage (as one does in model-Hamiltonian approaches) and without any support from previous experimental data, due to the DFT ab-initio nature. A connection to other theory tools is, however, very powerful. For example, symmetry analyses are often very helpful in this respect and Landau-theory approaches based on macroscopic order parameters (magnetization, ferroelectric polarization, etc.) are complementary to investigations oriented towards microscopic mechanisms.

DFT has many limitations though, mainly due to the restrictions allowing to simulate only periodic systems with “small” unit cells (i.e., of the order of hundred atoms per unit cell) and at zero Kelvin. A promising approach to move towards large-scale simulations at finite temperatures is the extension from first-principles to the so-called “second-principles” [30] via effective Hamiltonians, still keeping the DFT predictive power and accuracy. Effective Hamiltonian software packages based on underlying DFT calculations exist and are currently being developed, including a second-principles approach for lattice dynamics based on atomic potentials fitted on DFT.

Another severe limitation of DFT becomes evident when dealing with strong electronic correlations, intrinsically linked to the single-particle approach of basic DFT Kohn-Sham equations, which neglect the many-body nature often important in ferroics electronic systems. First-principles limits are often due to the approximation that is generally adopted for the DFT exchange-correlation potential: the electronic system is treated as equivalent to a homogeneous electron gas, i.e., behaving very differently from localized *3d* or *4f* states that often lie at the core of the ferroic phenomenology. Several techniques exist to overcome the electronic correlations problem, ranging from small modifications to DFT (such as local density approximation—LDA + U) to completely different approaches, such as dynamical mean field theory or quantum chemical methods (i.e., complete active space self-consistent field, CAS_SCF). Needless to say, the choice on which a theoretical approach is more appropriate for each scientific problem is often based on the tradeoff between computational cost and required accuracy.

In the aim of getting closer to technology, further steps are generally taken along the “theory pipeline” following ab-initio methods: (i) moving from discrete, atomistic models to continuum-based approaches (i.e., micromagnetic simulations allowing the study of magnetization dynamics on mesoscopic scales, for example, using MuMax3 [31]) and (ii) bridge the gap towards multiphysics simulation software packages (such as COMSOL [32]) and microelectronics design suites for realistic device modelling (SPICE [33]).

Within the current hype of artificial intelligence and big data and moving back to first-principles approaches, we note that DFT is not only valuable for the initial discovery of macroscopic mechanisms, but also for the subsequent optimization and design phases, which need to be carried out for the mechanism’s efficient exploitation. In this respect, we note that materials design is currently undergoing a revolution driven by the increasing importance of machine learning (ML) approaches as a boost for materials discovery. While at present ML for quantum matter is rather unshaped, it is likely to dramatically develop in the next decade, with all materials classes (including functional ferroics) expected to benefit from artificial intelligence. We observe that functional materials are particularly difficult to model on the basis of “intuition” or simple “rules of thumbs”, due to their inherent complexity. Furthermore, they often present cross-coupling phenomena (magnetostriction, piezoelectricity, magnetoelectricity), which make them very useful from the application point of view, but at the same time extremely difficult to predict and optimize without a complex theoretical machinery. The combination of first-principles simulations based on DFT (performed within a high-throughput approach in the aim of building medium-to-large datasets [https://materialsproject.org/, https://www.materialscloud.org/, http://aflowlib.org/, etc. (accessed on 23 June 2022)] to be analyzed with data-mining techniques) and of machine learning approaches (including those based on regression methods) seems, therefore, a powerful tool for future materials design.

### 3.2. Challenges in Material Synthesis and Advanced Characterization

Without the aim of a full coverage of these important aspects and inviting the reader to find detailed insights in many outstanding reviews and books recently published, in this section we focus on some representative examples. In these, we describe how the research challenges found in quantum materials reflect into advanced analytical and growth requirements that are well beyond the standard requests of chemical, structural and electronic structure investigation by conventional techniques. In perspective, two ‘transitions’ are needed in the mid-term future.

First, from ‘more-than-one’ to ‘all-in-one’ analytical capabilities. To reach this goal, in-operando techniques for measuring fundamental properties of materials while applying an external stimulus must move from proof-of-concept to engineered experimental setups, combining state-of-the-art analytical techniques with instruments able to measure functional properties. New instruments must be realized where resolution (spatial, energetic, atomic) is pushed to the limit, yet flexible enough to ‘commute’ to different regimes (e.g., from ms to fs in time, from eV to meV in energy, from micron to angstrom in space, from single layer to buried interface in depth). Second, a transition from the ‘observation age’ to the ‘control age’, long foreseen but still not reached [34], is necessary, where analytical, spectroscopic and functional tools must perform on materials, notably down to the single layer, with the same approach as industrial mechanical tools, i.e., bringing precision, accuracy and repeatability to the nanoscale limit.

#### 3.2.1. Dimensionality, Growth and Scalability

Recent years witnessed the emergence of a new family of quantum systems whose electronic and magnetic properties fall somewhere between those strictly 2D (e.g., graphene) and the ‘classic’ 3D ones (e.g., perovskite oxides, ferroics) [35,36,37].

Regarding spintronics, a common thread across several ‘*less than 3D*’ materials and their hetero-interfaces is that the strength of SOC in the solid is comparable to the energy scale of other electron-electron interactions, thus playing a relevant role in producing (new) spin-based phenomena [38]. Knowing that the interplay between SOC and (ferro-)magnetism is of paramount importance, their mutual relationship in 2D materials is less explored.

Furthermore, the number of intervening layers, the fabrication process and the role of a doping element are parameters to be controlled during synthesis and growth. The state-of-the-art mechanical exfoliation methods produce high quality monolayers, but this technique is not scalable, while scalability is one of the main prerequisites for applications. Despite considerable progress in their synthesis, the quality of many of the 2D magnetic systems has not yet reached that of the mechanically exfoliated flakes (e.g., mobility of mechanical cleaved samples if a factor 100 higher than the one in samples obtained by other methods). Common procedures and protocols for growth control and sample manipulation/transfer in UHV environments must be developed with the final goal of combining synthesis/growth and spectroscopy in the same experimental apparatus [39,40,41,42].

However, the high thermal and chemical stability of a three-dimensional (3D) crystal is essential before one can even contemplate the possibility of its 2D counterpart. Graphite has both, allowing graphene to exist under ambient conditions. For example, it is still challenging to obtain 2D monocrystals with uniform/homogeneous doping using traditional synthesis methods. Therefore, innovative non-equilibrium methods of synthesis need to be developed in order to achieve controllable doping in ‘less-than-3D’ lattices. In addition, the dynamics of local dopants are largely unexplored [43,44].

To address these challenges with the perspective of integrating deposition in an industrial fabrication process, the community should move to state-of-the-art, wafer-scale compatible synthesis techniques, exploiting the peculiarities and the combination of both physical vapor deposition techniques (sputtering, pulsed laser deposition, molecular beam epitaxy) and chemical techniques (chemical vapor deposition—CVD, plasma enhanced chemical vapor deposition—PECVD, atomic layer deposition—ALD) [45,46]. Recent results show that critical issues can be overcome, for instance, exploiting the peculiarities of van der Waals epitaxy with respect to conventional 3D MBE growth [47,48].

#### 3.2.2. Multi-Messenger Spectroscopic Tools

Practically all the critical features of synthesized layers mentioned above cannot be evaluated using a single characterization approach. The theoretical analysis of modifications induced on band structure by both doping and proximity, the controlled direct growth of the heterostructures (mono and few-layers thick) and the comparison with single crystal and/or exfoliated layers, are directly interconnected to the measurement of the electronic properties, of the spin texture and of domain structure when the system enters into a ferromagnetic/ferroelectric (FM/FE) or antiferromagnetic/antiferroelectric (AFM/AFE) phase, in a multiferroic phase. Moreover, both static and dynamic conditions need to be explored, implying a synergetic approach between lab-based and large-scale facility environments [49].

Here we outline a roadmap for relevant developments to be carried out in the coming years to provide relevant information on both the fundamental and functional properties of quantum materials in view of their deployment in microsystems.

Photoemission Spectroscopy (PES), and in particular, X-ray based PES based at large scale facilities such as synchrotron radiation sources, is one of the most powerful spectroscopic tools to investigate solid-state properties. PES is chemical-sensitive, the use of circular polarization allows the detection of long-range magnetic properties, while spin and angular resolved PES (S-ARPES) is the method of choice for the measurement of the band structure and Fermi surface of a solid [50,51,52].

In recent years, also boosted by the needed knowledge of spin texture in quantum materials, new and more efficient solutions have been implemented, with systems based on the exchange interaction between low-energy photoemitted electrons and magnetically ordered target surfaces, known as V-LEED (very low energy electron diffraction) detectors [41,53,54]. This has led to the increase in efficiency by a factor 100 and rendered spin-ARPES (S-ARPES) suitable for high-resolution experiments, with vectorial capabilities (i.e., measure of spin components along three orthogonal axes). Recently, multidimensional S-ARPES has been demonstrated, enabling measurements of band structure snapshots with spin contrast [55]. The implementation of efficient S-ARPES spectrometers suitable for fast acquisition of E(k) dispersion relationships with spin resolution would enable a much more efficient optimization of quantum materials, as well as the implementation of in-operando experiments, nowadays impossible due to the long acquisition time related to the fact that most S-ARPES spectrometers are sequentially sampling different wave vectors.

Surface and interface properties, however, cannot necessarily be predicted from those observed at larger or smaller scales and are not simply scalable from bulk or atomic properties. It is clear that an adequate knowledge on the surface/interface vs. bulk electronic properties is mandatory to foresee any real application, and a firm understanding of the mechanisms involved in these effects is required. A clear example of this difficulty can be found in the analysis of complex oxides. PES in the 10–500 eV of photoelectron kinetic energy range is extremely surface-sensitive, i.e., the probing depth is limited to only 5–10 Å [56]. This surface-sensitivity is, on the one hand, important for studying heterostructures with few layers, but on the other hand, could be problematic when, e.g., dealing with device-ready systems with buried interfaces and capping layers. The most direct way for increasing the probing depth in PES experiments is either to decrease or increase the excitation energy and, correspondingly, the photoelectron kinetic energy [57,58]. Paying the price of a limited energy resolution (<100 meV), hard X-ray photoemission spectroscopy (HAXPES) features a probing depth up to 20 nm for electron kinetic energy > 3 keV [59,60].

According to experiments and calculations, the probing depth in PES experiments should also steeply increase at very low electron energy (*E_k_* < 6 eV). In recent years, laser-based PES experiments in this energy range (hν~6 eV) demonstrate sub-meV energy resolution, access to high angular resolution (ARPES) and a probing depth up to some nm [61]. Recent findings [62] outline that the probing depth is actually strongly dependent upon the material and the proper selection of the excitation energy. It is thus mandatory to use a tunable energy source also in the low energy regime. The comparison of both low- and high-energy PES with true bulk sensitivity will be a unique feature to develop in future, allowing a full spectrum of information, including S-ARPES capabilities. Moving to functional properties, in-operando ARPES is an emerging powerful technique to investigate the effect of external stimuli on the electronic properties of novel materials [63,64,65].

In this respect, the development of novel experimental stations for studying the time-evolution of the band structure in prototypical devices allowing for the application of stimuli such as electric/magnetic fields, stress, temperature gradients, etc., necessarily involving thin top electrodes or capping layers, is of overwhelming relevance. This typically asks for the combination of HAXPES with pump-probe techniques and device characterization techniques for monitoring, in real-time, functional properties (magnetic, electric, ferroelectric, etc.) and how they are related to the band structure. So far, only a few proofs-of-concepts have been reported, while this kind of analysis should become a “routine” tool for a better understanding and engineering of quantum materials properties [66,67].

Of course, similar considerations also hold true for other experimental techniques based on synchrotron radiation or, in general, each kind of photon/electron—in/out technique, giving access to fundamental properties of matter, such as X-ray absorption techniques at large scale facilities (X-ray absorption spectroscopy, X-ray magnetic circular and linear dichroism, etc.), laterally resolved spectroscopies such as photoemission electron microscopy (PEEM), inverse photoemission and transmission electron microscopy.

### 3.3. Multiferroics for Non-Conventional Computing

As stated above, quantum ferroic materials naturally provide intriguing solutions for non-conventional computing strategies beyond Boolean computing.

#### 3.3.1. Neuromorphic Computing

In the von Neumann architecture, the physical separation between CPU and memory units requires a large amount of data to be transferred between them, and this necessarily limits the machine, both in time and in energy (von Neumann bottleneck). New brain-inspired computing paradigms (neuromorphic computing or memcomputing) allow overcoming this bottleneck, putting the whole burden of computation directly onto the memory [68]. This can be realized in practice by exploiting the physical properties of devices (memelements), which shows a certain degree of time non-locality (memory) in their response functions, such as memristors, memcapacitors and meminductors [69]. When involved in neural networks, they can outperform von Neumann machines in some tasks such as speech or pattern recognition and image segmentation. Nevertheless, none of the proposed memelements has been integrated in VLSI (very-large-scale integration) chips.

Materials that could enable information processes with state variables other than charge are very attractive, but often a fundamental issue in view of their exploitation is the poor compatibility with CMOS electronics. The magnetization of a ferromagnetic element involved in a spintronic device (e.g., a magnetic tunneling junction) can be used to implement a multilevel memory with plastic behavior, with different states encoded in the micromagnetic configuration of said elements which can be used as synaptic weights in neural networks (see Section 2). Some proof-of-concept papers exist showing a neuromorphic behavior related to the displacement of a domain wall in a ferromagnetic electrode upon application of current pulses which influence the magnetization via either spin transfer torque (STT) or spin orbit torque (SOT) [70,71,72]. Whatever the approach, however, the writing energy is not negligible due to the large current density needed both for STT and SOT (on the order of 10^6^–10^7^ A cm^−2^). Possible solutions come from quantum materials, especially Rashba systems and topological insulators displaying spin-momentum locking and much larger values of charge-to-spin conversion efficiency with respect to heavy metals (Pt, Ta, W). Nevertheless, so far, a convincing demonstration of neuromorphic devices exploiting topological insulators or Rashba systems for SOT-based magnetic tunneling junctions is still missing. On the other hand, ferroelectric (FE) polarization (P) is a very intriguing option as a state variable because it: (i) is purely electric, (ii) can be altered via low power voltage pulses and (iii) ensures long-term endurance [20].

#### 3.3.2. Wave Computing

The potential of optics for signal processing is at the basis of the great success of photonics in telecommunications. A complex operation, such as a Fourier transform of a 2D signal, is easily operated by a simple lens; filtering and spectrum analysis can be implemented using optical elements which exploit typical wave phenomena such as diffraction and interference. This was the basis of the excitement of the scientific community in the 1960s, when such concepts were demonstrated with discrete optical components, as well as at the beginning of this century, with the development of integrated optics and silicon photonics [73]. The integration on a microchip of Fourier optics represents a sort of “holy grail” for photonics, but so far, the integration of photonics in microelectronic systems is mainly related to optical interconnects, not to real signal processing which is always performed by CMOS-based electronics in the digital world. The main reason being that, so far, integrated optics platforms are essentially operating in an open-loop and display poor reconfigurability. To give an example, the modulation of the phase shift in one arm of a Mach–Zender interferometer (a basic component in integrated optics) is still achieved via thermal modification of the Si refractive index, with the obvious limitations in terms of energy consumption and operation speed (switching times, bandwidth). To overcome these limitations, quantum materials can make the difference. Piezoelectric and birefringent materials can be used to modify the light propagation in integrated waveguides, either by mechanically displacing waveguides on a chip in such a way as to modify the coupling via the evanescent waves, or by changing the complex refractive index of the waveguide (or its cladding) upon application of electric pulses to ferroic materials.

Beyond that, quantum materials can support the propagation of quasi particles, other than photons, with peculiar properties in terms of wave dispersion and nonlinearity. A specific example is that of magnons or spin waves (SWs), i.e., the elementary excitation of a spin lattice which can be visualized as a wave resulting from the local precession of the magnetization around an effective field with a phase shift which increases along the spin chain. The definition above strictly applies to exchange magnons, i.e., excited states of a spin lattice described by the Heisemberg Hamiltonian, leading to spin waves with short wavelength (up to tenths-hundreds of nm). Larger wavelength spin waves exist (in the range of microns), the so-called magnetostatic SWs, which are mainly due to the dipolar interaction between neighbor spins and represents a specific solution of the Maxwell equation in ferromagnetic media, whose dynamics are described by the Landau–Lifshitz–Gilbert equation [74]. Whatever the nature of spin waves (exchange or dipolar), they display all the phenomena typical of waves (polarization, interference, diffraction, etc.), and many integrated devices have been proposed for signal processing or computing [75,76]. Furthermore, the peculiar dispersion relation of spin waves with respect to electromagnetic waves in free space offers intriguing opportunities for device miniaturization. To give some perspective, while the free-space wavelength for microwaves in the GHz range is in the order of cm, for the same frequency exchange, SWs can display wavelengths below 100 nm, thus paving the route to integrated magnonic devices for RF signal processing.

This said, many issues are still unsolved in the field of magnonics. First, typical insertion losses for full RF devices with electric input and output are currently too high (on the order of 10^−4^) to envisage practical application of the proposed devices. This is not due to power dissipation during signal processing with SWs, as SW propagation does not imply joule dissipation and the attenuation length due to magnon-lattice interaction is in the order of mm for low-damping materials such as yttrium iron garnet (YIG). Instead, it depends on the low conversion efficiency between input and output electromagnetic waves and SWs, especially when this is accomplished via conventional RF antennas. New strategies have been proposed, such as the use of integrated patterned spin textures as antennas/receivers, spin hall or point contact nano-oscillators for SW generation, magnetoelectric coupling for SW excitation/detection, etc. [77,78,79]. For all proposed solutions, quantum materials are very promising to overcome current limitations and improve the overall device energy efficiency. In particular, topological insulators and Rashba systems represent a viable alternative to widely-used heavy metals for the excitation of SWs via charge-to-spin conversion with an efficiency gain of a factor of 100 [80,81]. Multiferroic quantum materials displaying spin-momentum locking (GeTe, SnTe, etc.) offer the additional advantage of the non-volatility of the ferroic order which could be used for a new generation of reconfigurable magnonic devices.

#### 3.3.3. Quantum Computing

A single-spin system is the prototypical qubit. Nevertheless, the implementation of quantum computing with single donor atoms in Si [82] or in quantum dots [83] is still limited to only to a few qubits in prototypes which are far away from integration. Among many others, a critical bottleneck is the difficulty to integrate a means to produce the variable magnetic fields needed for quantum operations. In this framework, multiferroics could represent an interesting opportunity to use magnetoelectric coupling for the electric control of the stray field produced by peculiar micromagnetic configurations in ferromagnetic films instead of that produced by external magnets.

However, apart from single donors and quantum dots in semiconductors, there is an emerging branch of quantum computing which tries to use spin waves to couple magnetic qubits [76]. Of course, in this perspective all considerations of Section 3.3.2 on the potential of quantum materials for magnonics can be immediately transferred to the field of quantum computing.

Finally, let us mention the opportunity of using ferroic materials to implement, on-chip, the electrical reconfiguration of integrated optical circuits for quantum computing with a single photon. The solution of the abovementioned issue related to the poor reconfigurability of integrated photonics with ferroic materials would pave the way to the development of integrated quantum computing with photons.

### 3.4. Device Engineering and Exploitation

There is not a unique recipe for bridging from a proof-of-concept device to industrial exploitation and consequent high socio-economic impact, but some prerequisites can be clearly pointed out. *First:* materials should be non-toxic, abundant and with a sustainable supply chain. Researchers should start considering, not only the performance of their proof-of-concept devices, but also figure out their life cycle assessment (LCA) which is becoming of overwhelming importance also for industries. Even though we are late in recognizing this, the impact of each product on climate changes and world pollution must be seriously considered and will become a relevant factor in the selection of forthcoming technology platforms. *Second*: fabrication processes must be wafer-scale and CMOS-compatible. New fabrications are definitely working on 300 mm wafers and any novel material or fabrication processes must be upscalable to this size. On the other hand, whatever the application (microelectronics, photonics, micromechanical systems, etc.), emergent devices will require the integration with CMOS electronics for their control and communication with the web. Thus, all steps of fabrication processes should respect CMOS requirements in terms of chemical contamination, thermal budget, etc. *Third:* curiosity-driven research is of fundamental relevance, but for mid-term exploitation and “fast technology transfer”, research projects going from low- to intermediate-technology readiness level (TRL), involving theoretical and experimental researchers from both academia and industry, are needed to orient the deployment of exciting physical concepts towards realistic applications.

## 4. The Prototypical Case of GeTe

During the last 8 years, the authors have applied the methodology proposed in Section 3 to a peculiar ferroic material with unconventional properties, i.e., GeTe [84], a widely investigated and CMOS-compatible [85] phase-change semiconductor which is also ferroelectric and has become the father-compound of a new class of semiconductors which we have introduced: ferroelectric Rashba semiconductors (FERSC) [86]. Being ferroelectric, FERSC lack inversion symmetry. As such, in the presence of spin-orbit coupling which breaks spin-degeneracy when combined with non-centrosymmetry, they show a k-dependent spin-splitting in some regions of the Brillouin zone owing to the (bulk) Rashba-effect. What is relevant is, that by means of pioneering DFT calculations (see Figure 2), the spin direction in each sub-band (arrows in Figure 2a) was found to be fully reversed upon reversal of the ferroelectric (FE) polarization. This ensures the permanent, non-volatile electric control of spin degrees of freedom, in principle, allowing the long-sought integration of spintronics with ferroelectricity in CMOS-compatible materials. The GeTe band structure around the Z-point is reported in Figure 2 along the ZA and ZU directions (see the left column), showing a peculiar Rashba-like splitting which is particularly evident in the valence band. Noteworthy, as schematically depicted in Figure 2a, spin-momentum locking appears in simulated isoenergy cuts of Figure 2c, with the spins circulating clockwise (counterclockwise) in the outer (inner) band for outwards polarization (P). Reversing the ferroelectric polarization (Figure 2d) causes a reversal of the spin circulation direction.

Starting from the original DFT calculations, S-ARPES experiments have been crucial to demonstrate the concept of FERSC materials, proving the presence of a giant Rashba-effect in ferroelectric GeTe, linked to the ferroelectric polarization [87]. Figure 3 and Figure 4 report the main result of ref. [88], i.e., the fact that the sense of circulation of spins measured by S-ARPES changes for the Te-rich surfaces (P outwards) and Ge-rich surfaces (P inwards). By exploiting the natural tendency to stabilize a specific polarization state as a function of the chemical composition of the outermost layer, we have been able to investigate the spin texture for opposite P without gating, which is difficult to implement for ARPES investigations with UV radiation. In particular, Figure 3 refers to the case of a Te-rich GeTe(111) surface, with outward ferroelectric polarization. By measuring spin-polarized spectra at opposite wave vectors (k_1_ and −k_1_) where bulk Rashba bands B_1_ and B_2_ are well distinguishable and both below the Fermi level (see Figure 3a), a spin texture coherent with the theoretical prediction was found. As a matter of fact, with reference to the spin quantization axis of Figure 3, the spin-polarization of spectra at +k_1_ (see Figure 3b,c) is opposite to that of spectra at −k_1_ (Figure 3d,e). This demonstrates that spins display an opposite sense of circulation in the inner and outer Rashba bands, as depicted in the isoenergy cuts at 0.18 and 0.5 eV of Figure 3f,g, where red (blue) arrows indicate the spin orientation in outer (inner) bands crossed at that energy.

By slightly increasing the temperature during the in-vacuum annealing used to de-cap the GeTe sample from the Te protecting layer, a Ge-rich GeTe surface with inwards ferroelectric polarization can be obtained. A similar S-ARPES analysis has been performed on said surface and results are summarized in Figure 4. Apart from the slightly different choice of the two opposite wave vectors (k_2_ and −k_2_) where the spin-dependent analysis has been carried out, spin-polarized spectra reported in Figure 4b–e reveal a clear inversion of the spin-polarization when moving at opposite k, as summarized in the isoenergy cuts of Figure 4f,g. The great difference with Figure 3 being that the sense of circulation is now reverted due to the change of P: counterclockwise in the outer band and clockwise in the inner one. Together, Figure 3 and Figure 4 provide a clear experimental demonstration that the ferroelectric polarization determines the sense of rotation of the spins in Rashba bands displaying spin-momentum locking.

After proving the link between ferroelectric polarization and spin texture [87,88], we demonstrated the macroscopic switchability and endurance of the FE polarization by electric gating. Furthermore, we provided evidence for the ferroelectric control of the spin-to-charge conversion, i.e., the fact that the sign of the spin-hall conductivity changes upon reversal of the ferroelectric polarization in experiments of spin-pumping on gated Fe/GeTe structures [23,89].

The GeTe story is a prototypical example of the systematic approach we believe is needed to move from materials to microelectronic systems. Since the very beginning, we have been attracted by the “simplicity” of the compound (only two elements), its high Curie temperature (which means room temperature operation), the high level of multifunctionality and the fact that it was already CMOS-compatible, due to its deployment in the framework of phase-change memories. Rather than choosing “exotic” compounds, we first selected a quantum material holding a high potential for applications and then we started a systematic investigation of fundamental properties. Despite the initial skepticism of the scientific community, especially concerning the possibility of reliable writing and reading of ferroelectric domains in a semiconductor material, the soundness of our results has attracted the attention of the community: the seminal paper by D. Di Sante et al. [84] is receiving a high number of citations and many groups worldwide are turning their attention towards the fascinating properties of GeTe. On the other hand, the deep understanding of GeTe’s fundamental properties that we have accumulated over the last 8 years constitutes the solid basis to move to the next step (see Figure 5): the development of computing devices exploiting ferroelectricity as a variable of state and spin to charge conversion for reading, in analogy with the recent proposal by INTEL [19] of the MESO structure. In the MESO, the memory state is stored via magneto-electric coupling in a ferromagnetic electrode and read by spin-to-charge conversion of the spin-current injected from the ferromagnetic electrode into a topological insulator or Rashba system. Our concept device, called FESO (ferroelectric spin orbit device), has a reduced degree of complexity and possibly lower power consumption. In fact, at variance with the INTEL proposal, we need only a single material (GeTe) which implements both the writing (ferroelectric switching by gating), the memory functionality (ferroelectric variable of state) and the reading (by spin-to-charge conversion due to inverse spin-hall or Edelstein–Rashba effects). Interestingly enough, due to the limited size of ferroelectric domains, there are no fundamental limitations to the FESO downscaling to a few tenths of nanometers [23].

Finally, we note that dealing with a material which is already available in semiconductors fabs has also been a key factor in relation to potential end-users of this technology. 

## Figures and Tables

**Figure 1 materials-15-04478-f001:**
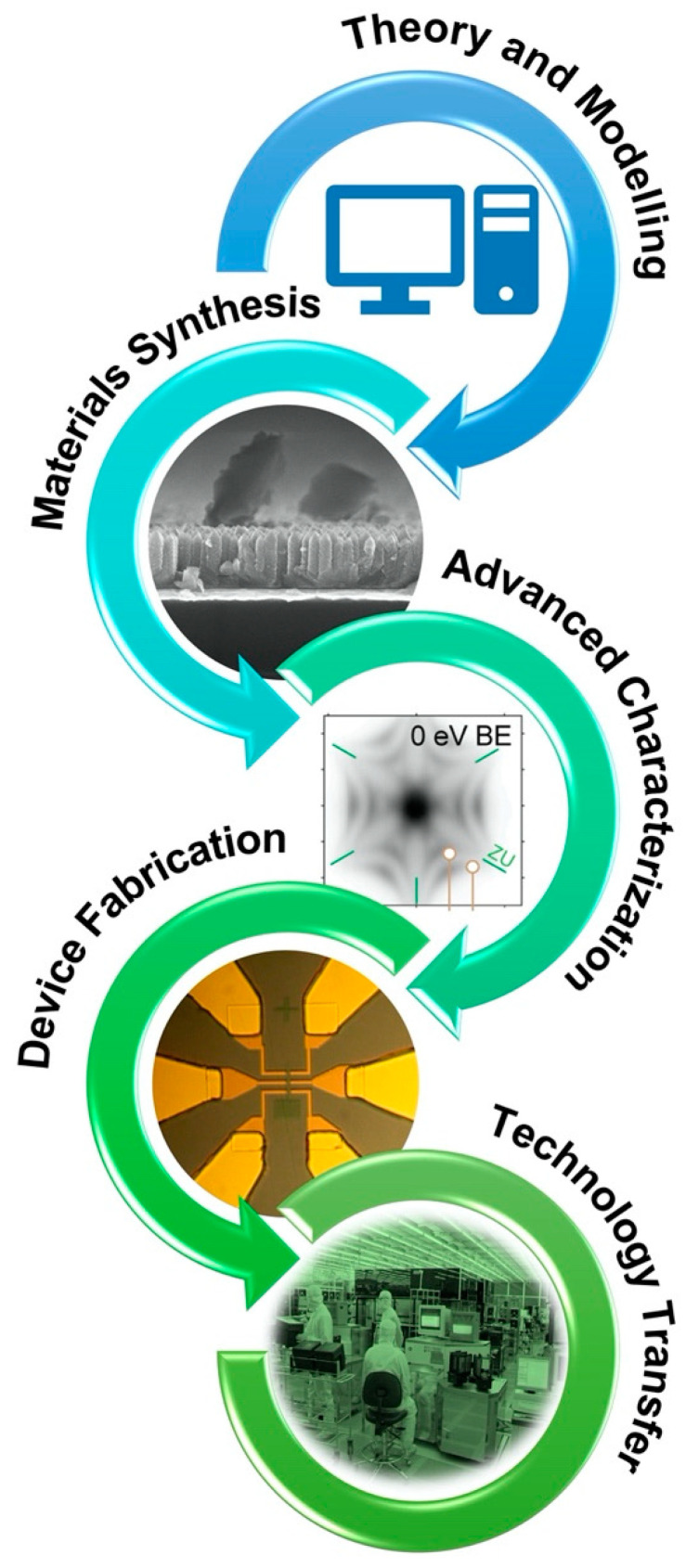
Workflow for a roadmap towards industrial exploitation of quantum materials.

**Figure 2 materials-15-04478-f002:**
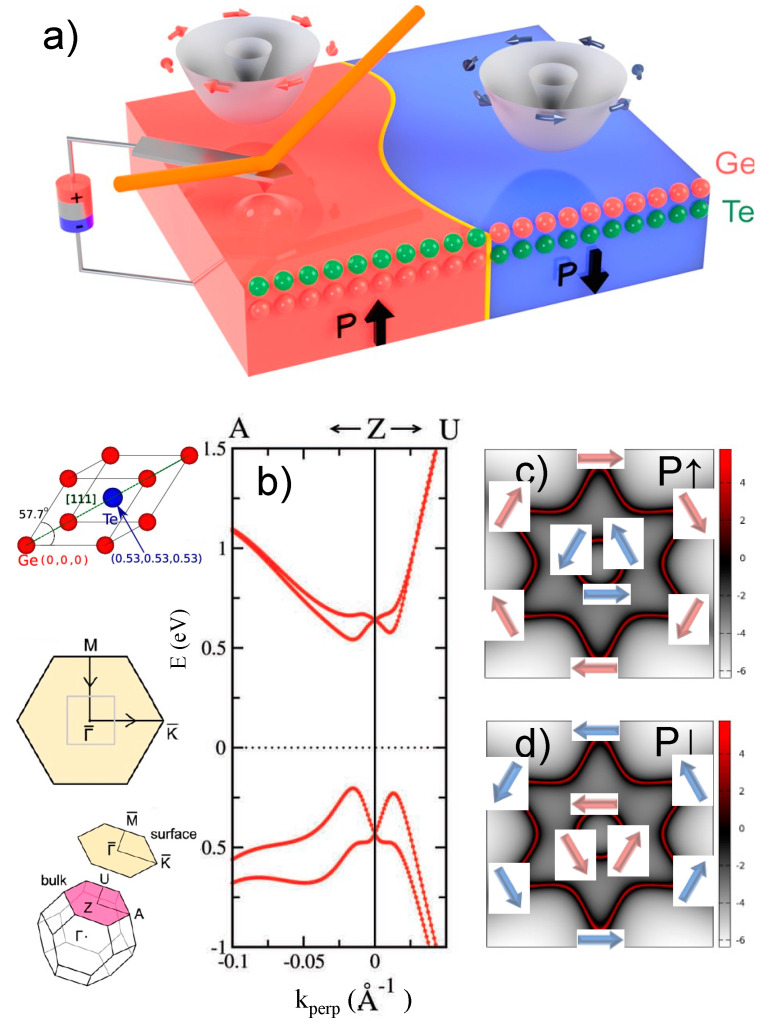
(**a**) FERSC concept, based on the link between polarization and spin texture: for polarization pointing “up” (“down”), i.e., P**↑** (P**↓**), the spin expectation value in the outer branch of the Rashba bands is circulating clockwise (counterclockwise), whereas in the inner branch it is circulating counterclockwise (clockwise). Reprinted with permission from Ref. [88]. Copyright © 2022 American Chemical Society. (**b**) GeTe band structure, calculated via DFT, in the two directions Z-A and Z-U for the highest valence band and lowest conduction band. In the left column, the GeTe rhombohedral crystal structure is shown along with the bulk Brillouin zone and its projection on the experimentally most stable surface. A squared zoom is also shown around the bulk Z-point (corresponding to the *Γ* point on the surface), lying in a plane perpendicular to the polarization direction (in reciprocal space). (**c**) isoenergy cut at around −0.5 eV for polarization pointing “up”, plotted in the abovementioned square around the Z-point. The arrows denote the spintexture of the inner and outer Rashba bands. (**d**) same as (**c**) but for polarization pointing “down”. COPYRIGHT: panel (**a**). Reprinted with permission from Ref. [84]. Copyright © 2022 John Wiley and Sons, Inc.

**Figure 3 materials-15-04478-f003:**
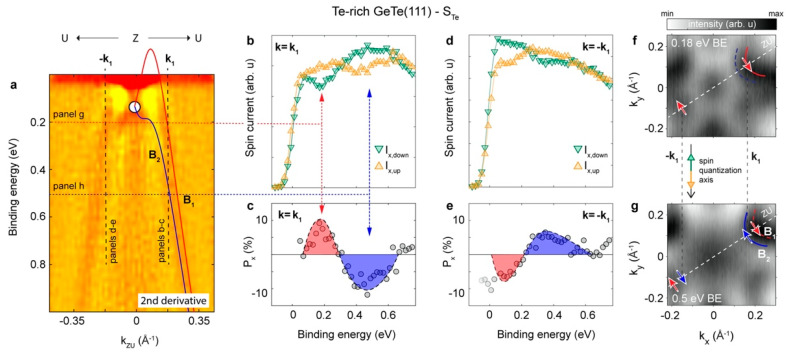
Spin-resolved ARPES for the Te-rich sample (P outwards). (**a**) Calculated bulk bands (solid line) along ZU (k_y_) superposed to the 2nd derivative of the measured band dispersion. (**b**,**c**) Spin-polarized spectra and spin asymmetry at fixed wave vector k_1_ indicated in panel a, with the two peaks related to bulk Rashba states at k_1_ from B_1_ and B_2_ bands (panel a). (**d**,**e**) Spin-polarized spectra and spin asymmetry at opposite wave vector −k_1_. (**f**,**g**) Constant energy maps at 0.18 and 0.5 eV BE, corresponding to the energy of bulk bands B_1_ and B_2_ at k_1_. Blue and red arrows indicate the sense of circulation of spins: clockwise in the outer band and counterclockwise in the inner one. Reprinted with permission from Ref. [88]. Copyright © 2022 American Chemical Society.

**Figure 4 materials-15-04478-f004:**
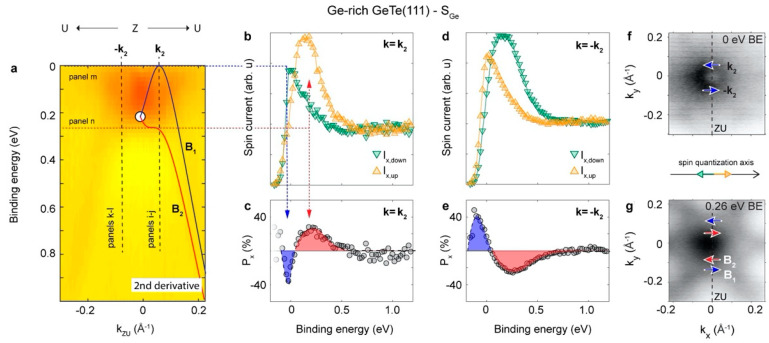
Spin-resolved ARPES from a Ge-rich sample (P inwards). (**a**) Calculated bulk bands (solid line) along ZU (k_y_) superposed to the 2nd derivative of the measured band dispersion. (**b**,**c**) Spin-polarized spectra and spin asymmetry at fixed wave vector k_2_ indicated in panel a, with the two peaks related to bulk Rashba states at k_2_ from B_1_ and B_2_ bands (panel a). (**d**,**e**) Spin-polarized spectra and spin asymmetry at opposite wave vector −k_2_. (**f**,**g**) Constant energy maps at 0 and 0.26 eV BE, corresponding to the energy of bulk bands B_1_ and B_2_ at k_2_. The sense of circulation of spins is opposite to that found for the Te-rich sample (see Figure 3): counterclockwise in the outer band and clockwise in the inner band (**g**), thus demonstrating that the spin texture is controlled by the ferroelectric polarization. Blue and red arrows indicate the sense of circulation of spins: counterclockwise in the outer band and clockwise in the inner one. Reprinted with permission from Ref. [88]. Copyright © 2022 American Chemical Society.

**Figure 5 materials-15-04478-f005:**
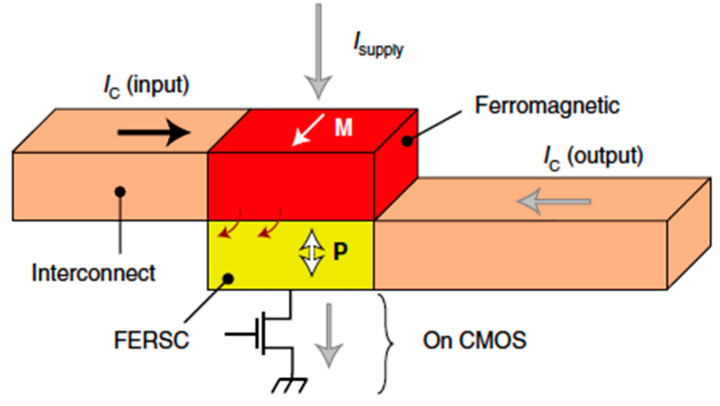
Concept of a spin–orbit transduction device with FERSC exploiting the ferroelectric control of the SCC. A ferromagnetic layer with fixed magnetization injects a spin-polarized current into the active FERSC element when the supply current (I_supply_) is on. An output charge current is generated by inverse spin-hall effect with a sign related to the polarization state. Reprinted with permission from Ref. [23]. Copyright © 2022 Springer Nature.

**Table 1 materials-15-04478-t001:** Comparison between digital (DRAM) and analog (ferroic-based) strategies for storing the synaptic weight to be used in a neural network for patterning recognition.

Figure of Merit	DRAM (32 bit)	FERSC (FESO)	FE 2DEG (MESO)	TI (SOT MTJs)
Wafer size	12”	Small samples *	Small samples	Small samples
Access energy	0.64 nJ	10 fJ	100 pJ	1 pJ
Writing/reading time	10 ns	<250 ns	100 ns	100 ms
Retention time	64 s	>1 h	>1 h	years
Operating temperature	RT	RT	<50 K	RT

* typically 10 × 10 mm^2^ or smaller.

## Data Availability

Data reported in the present paper are taken from previously published papers by the authors.

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
