# Peer review of "From Quantum Materials to Microsystems"

_materials, 2022, doi:10.3390/ma15134478_

Round 1

Reviewer 1 Report

Dear authors, 

Even though the type of paper (i.e. Perspective), i find it very general, where the state-of-the-art is not very clear, but also, reporting in a very plain way, what current literature states (including some plagiarism therein). 

Therefore, I would suggest greatly revising it by giving clear comparative tables, removing any "plagiarism" and citations from wikipedia and not valuable sources, and try to resubmit. 

In the attached pdf, I am giving you some of the main problems arises with your manuscript. 

Author Response

We thank the referee for reviewing our manuscript. According to her/his suggestions we tried to improve the paper soundness by avoiding any excess of quote or "plagiarism". To put our discussion on a more soldi basis, we introduced a comparative table to benchmark ferroic-based solutions to current CMOS technology in case of nonconventional computing.

Please refer to the attached file for a point to point reply to the specific comments.

Reviewer 2 Report

In this paper, the authors analyse the ferroic materials to outline the roadmap towards a ferroics’ realistic exploitation in microelectronic devices. The manuscript core contains four parts: theory and modeling, to unveil new microscopic mechanisms and emergent phenomena; material synthesis and advanced characterization; device design, fabrication and characterization; technology transfer, where the authors describe how to apply the technology of quantum materials. Conclusions are in the manuscript line.

Paper subject is interesting and falls within the journal topic. In general, the manuscript is well written, and it is supported by an impressive references list, including the authors contributions to the field.

 However, there some questions should be addressed:

1.      1. Sometimes the authors style is not so appropriate to a such journal. For instance, the impersonal style should be adopted. Some expressions are too ‘metaphoric’: “lucky” discoveries. Other times the style is didactic: ‘‘From the semiconductor lesson we learnt that’’. Please, revise the style throughout the manuscript.

2.      2. Comments to figure should be included in the main text, not in the figure legend.

Author Response

We thank there Reviewer 2 for appreciating our manuscript. Please find in the attached file the point-by-point reply to specific issues.

Reviewer 3 Report

Please see the enclosed file.

Author Response

We thank the Referee for reviewing our manuscript. Please find in attachment the point-to-point reply to raised issues.

Reviewer 4 Report

I have reviewed the manuscript “From quantum materials to microsystems” submitted to “Materials” for publication. In this study, authors have evaluated a specific class of quantum materials. This is a well-designed and well conducted study and the manuscript fits well within the scope of the journal; it needs some major improvements; there are a few suggestions that authors may consider to improve it further:

The use of English language is reasonable, however, there are a number of punctuation and grammatical errors; that should be corrected and rephrased using academic English for a better flow of text for reader.

Title: the format of the title should be corrected.

Authors should make sure that all the abbreviations are defined at their first appearance in the text and use abbreviations afterword.

Introduction is very comprehensive and detailing all the background information and rationale of the study.

For the line 31: Wikipedia quotes: “Quantum….” It is suggested to cite the original and an authentic reference instead of the Wikipedia.

Line 31-41: please cite suitable reference for this information.

Aim of the study is well described in the introduction. However should be clearly mentioned in the abstract section too. I suggest not to have a separate section for aims in the introduction for the scientific paper.

For figure 2: please confirm for having the permission for reusing this image.

There are several statement that is not mentioning any citation. Please make sure that appropriate references have been cited to support the provided information throughout the manuscript.

Authors should discuss the limitations of the study and describe future directions/recommendations.

Conclusion section should be precise and to the point; so it is suggested to remove or modify the presented text. 

Author Response

We thank the Reviewer for appreciating the manuscript. Please find in attachment the point by point response to specific comments.

Round 2

Reviewer 1 Report

Dear authors,

This revised version of your manuscript it is now well improved and ready for publication in the Journal of Materials. 

Looking forward to see it online. 

Author Response

We thank the reviewer for appreciating the improvement of the new version of the manuscript. According to the the Editor suggestions we are providing updated graphical files for Figures 1 and 3, so as to move to publication.

On behalf of all the authors

Riccardo Bertacco

Reviewer 4 Report

Thank you for revising the manuscript.

Author Response

We thank the referee for this second report. We are now providing updated figures for publication.

On behalf of all the authors

Riccardo Bertacco